# Exploring the use of seasonal forecasts to adapt flood insurance premiums

Viet Dung Nguyen[1], Jeroen Aerts[2], Max Tesselaar[2], Wouter Boutzen[2], Heidi Kreibich[1], Lorenzo Alfieri[3], Bruno Merz[1,4]

[1] Section Hydrology, Helmholtz Centre Potsdam, GFZ German Research Centre for Geosciences, 14473 Potsdam, Germany

[2] Institute for Environmental Studies, VU University, Amsterdam, The Netherlands

[3] CIMA Research Foundation, Savona, Italy

[4] Institute for Environmental Sciences and Geography, University of Potsdam, 14476 Potsdam, Germany

*Correspondence to*: Viet Dung Nguyen (viet.dung.nguyen@gfz-potsdam.de)

**Abstract.** Insurance is an important element of flood risk management providing financial compensation after disastrous losses. In a competitive market, insurers need to base their premiums on the most accurate risk estimation. To this end, (recent) historic loss data is used. However, climate variability can substantially affect flood risk, and anticipating such variations could provide a competitive gain. For instance, for a year with higher flood probabilities, the insurer might raise premiums to hedge against the increased risk or communicate the increased risk to policy-holders encouraging risk-reduction measures. In this explorative study, we investigate how seasonal flood forecasts could be used to adapt flood insurance premiums on an annual basis. In an application for Germany, we apply a forecasting method that predicts winter flood probability distributions conditioned on the catchment wetness in the season ahead. The deviation from the long-term is used to calculate deviations in Expected Annual Damage which serve as input into an insurance model to compute deviations in household insurance premiums for the upcoming year. Our study suggests that the temporal variations in flood probabilities are substantial, leading to significant variations in flood risk and premiums. As our models are based on a range of assumptions and as the skill of seasonal flood forecasts is still limited, particularly in Central Europe, our study is seen as first demonstration of how seasonal forecasting could be combined with risk and insurance models to inform the (re-)insurance sector about upcoming changes in risk.

## 1 Introduction

Extreme floods cause tens of billions of damages every year around the globe. Climate change will further amplify these losses due to more frequent and intense rainfall (Merz et al., 2021). Flood risk adaptation measures, such as flood protection by levees, are required to reduce (future) risk but residual risk will remain. To cover for residual risk, insurers play an important role by providing financial compensation to communities and companies to rebuild after a disastrous flood event. The availability of insurance dampens the impacts from flood disasters on livelihoods and the economy and accelerates recovery (e.g. Botzen and van den Bergh, 2008). However, the insurance industry faces several challenges regarding the coverage of natural disasters such as floods, including fat-tailed and dependent risk distributions, which are difficult to quantify (Kousky, 2019). The challenge concerning the quantification of natural disaster risks is that their estimation often depends on empirical evidence of natural disasters, which is scarcely available in many regions.

Many natural hazard insurance products that apply an indemnity insurance mechanism determine premiums annually, using empirical evidence of past events (Meier and Outreville, 2006). If an insurance market is competitive,

insurers will strive to apply the most accurate forecasting models and, thereby, gain a competitive advantage over other insurers. For example, an insurer may capture a larger market share when it can better predict below-average risk in an upcoming year than competitors, which allows it to charge lower premiums. On the other hand, the insurer that can better forecast a year with higher damages caused by natural hazards can also more effectively anticipate its business strategy than competitors without this knowledge. Such anticipation may include raising annual premiums, which would give the insurer a more comfortable financial position to cover damages (e.g. Hudson et al., 2019), but it may also include communicating the upcoming hazards to policy-holders and encouraging risk-reduction measures (De Ruig et al., 2022). If the latter strategy is done effectively, the insurer may not have to lose its market share by increasing premiums, while still maintaining a healthy financial position.

For the insurance and re-insurance sector, prior knowledge of possible (de-)increases in flood losses could assist in more reliable pricing of flood risk and related insurance premiums. An option are seasonal climate forecasts based on coupled ocean-atmosphere models, for instance the long-range forecasts of ECMWF (ECMWF, 2023), which can be used as forcing to hydrological models to obtain flood forecasts for the next months (Bennett et al., 2016). Another approach builds on statistical models linking next-season flood behaviour to the current climate and catchment state (Kwon et al., 2008). The idea of using seasonal forecasts to predict weather extremes over several weeks to months is not new, but their potential for the (re-)insurance sector has hardly been explored so far. Dlugolecki (2000) has addressed the potential of such forecasts for adjusting underwriting policies. More recent literature has explored the use of seasonal forecasts related to index based crop insurance (e.g. Cabrera et al., 2006, 2009). Carriquiry and Osgood (2012) demonstrate how to couple the provision of loans for covering cropping losses in an index insurance scheme to seasonal forecasts. Their results show that reliable seasonal forecasts may trigger the loan-insurance contract such that it may "substantially benefit participating farmers". Contrary, Majid Shafiee-Jood et al. (2014) demonstrate that seasonal forecasts cannot adequately predict extreme droughts and that the use of forecast information could actually worsen the potential benefit for farmers. This negative effect can be caused by the forecast uncertainty but also because of the issue of adverse selection: Only farmers who have forecasting information and face a drought take insurance while others do not. Maynard (2016) developed a synthetic hurricane loss forecasting model to assess the effect of including forecasting information in the pricing of insurance. This research indicates that while forecasting information may have benefits for policy-holders in producing on average lower premiums, it could have negative consequences for insurers: predicting higher losses would require companies to hold more capital, which can be more costly.

Seasonal forecasts are mostly applied to index based crop insurance studies and research shows that forecast uncertainty plays a decisive role for their usefulness in an insurance context. Against this background it is important to note that seasonal climate forecasts from dynamical or data-driven (statistical and AI-based) models have seen great advancements in recent years. For dynamical forecasting models, improvements in predictive skill have mainly benefited from improved physical process representation and model initialization, the emergence of ensemble forecasts representing uncertainty and computing advances (Jia et al., 2015, Bauer et al., 2016, Zhang et al., 2023). Data-driven seasonal forecasting has benefited from the improved estimation of initial hydrologic conditions and incorporation of climate information, as well as the advent of large datasets and AI-based forecasting algorithms capable of handling nonlinear relationships (Mendoza et al., 2017, Huang et al., 2020). Meanwhile, the majority of dams in the United States, for example, rely on seasonal forecasts of reservoir inflows to decide about water release (Turner et al., 2020).

The skill of seasonal forecasts results from slowly evolving earth surface boundary conditions, primarily sea surface temperature but also soil moisture, snow cover, and sea ice, and their impacts on weather (Robertson et al., 2020). Globally, ENSO (El Niño-Southern Oscillation) is the most prominent source of seasonal variability and can be forecasted with lead times of several months. ENSO and other climate modes (e.g. North Atlantic Oscillation) are able to modify the intensity of floods (Ward et al., 2014, Kundzewicz et al., 2019). Such climate-flood linkages lead to periods of above or below average flood peaks and losses (Zanardo et al., 2019). In addition to such climate teleconnections, floods are affected by the catchment wetness in the season ahead (Aguilar et al., 2017, Merz et al., 2018). Substantial influences of season-ahead catchment wetness on flood peaks are expected in regions where the water stored in the catchment is slowly released, for instance, in catchments with high contributions of groundwater to flood runoff. Another mechanism for a substantial link between season-ahead catchment state and flooding are regions where flood generation is linked to snowmelt and thus to snow accumulation in the season ahead.

Seasonal streamflow forecasts are successfully used to inform water resources management, such as securing navigation (Meißner et al., 2017), and managing reservoirs (Turner et al., 2017) and irrigation (Apel et al., 2019). However, most applications are related to low flow and drought management, while its use for flood management is largely unexplored (Arnal et al., 2018). In case, flood characteristics can be forecasted with lead times of several weeks to months, early alerts can be issued complementing the more widespread short-term flood forecasts. Such early alerts would allow additional flood mitigation measures, such as the pre-release of water from reservoirs to hedge against the increased probability of flooding in the coming season.

Given the progress in recent years in seasonal flood forecasting methods, we explore in this paper how forecasting data may influence evaluating and improving flood insurance schemes. In an application for Germany, we use a statistical forecast model that predicts winter flood peaks based on autumn precipitation conditions a few months earlier. The predicted deviation in flood peaks from the long-term mean is used to simulate the deviation in flood risk, which is subsequently used as input to an established flood insurance model to compute adjustments to insurance premiums for the upcoming year. We are aware of the uncertainties related to both seasonal flood forecasting and assumptions in our explorative study. Hence, we view this research as a demonstration of how seasonal forecasting information could be used for the insurance sector and as an impulse for investigating the opportunities of this approach in further studies.

**2 Case study area Germany**

Germany (Fig. 1) has seen several catastrophic river floods in the recent decades (Kreibich et al., 2017). For instance, widespread precipitation on saturated soils led to the 1993 Christmas flood in the middle and lower Rhine, with inundation in three federal states. Other examples are the events in 2002 and 2013 with disastrous damages in the Danube and Elbe catchments. Flood seasonality shows a spatial pattern across Germany with a winter-dominated flood regime in western, central, northern and eastern parts. While in the first two regions most of the annual flood peaks occur in winter, the latter two regions show an increased fraction of spring and summer floods (Burton & Thieken, 2009). Only a relatively small region in southern Germany is dominated by summer floods. For the sake of simplicity, we limit our analysis to the dominant flood season, i.e. the winter season, and develop a forecasting model that predicts the flood probability distribution in the upcoming winter season using climate information in the preceding autumn.

Flood insurance coverage is available in Germany since 1991 as a supplementary contract to building or contents insurance (Surminski and Thieken 2017, Seifert et al. 2013), which bundles flood risks with other natural hazard risks like earthquakes or avalanches. Traditionally, flood insurance penetration rates were rather low in Germany, however, a number of measures have contributed to the continuous increase of flood insurance penetration to 49% for residential buildings and 32% for household contents (GDV, 2022). For instance, the flood hazard zoning system (ZÜRS) was established by the German Insurance Association (GDV) in 2001 and is since then used to assess the insurability of properties (Falkenhagen, 2005). The special situation in the former German Democratic Republic (GDR) in East Germany, where flood damage was covered by household insurance, is no longer visible in the flood insurance penetration rates today. In contrast, the exceptionally high flood insurance penetration of 94% in the federal state of Baden-Württemberg (GDV, 2022) is a consequence of the fact that flood loss compensation was included in compulsory building insurance until 1994 in this federal state (Surminski and Thieken 2017, Seifert et al. 2013). Due to EU regulations this monopoly insurance had to be abandoned. Calls for a compulsory flood insurance scheme in Germany after the floods in 2002 and 2013 have been rejected (Schwarze and Wagner, 2004).

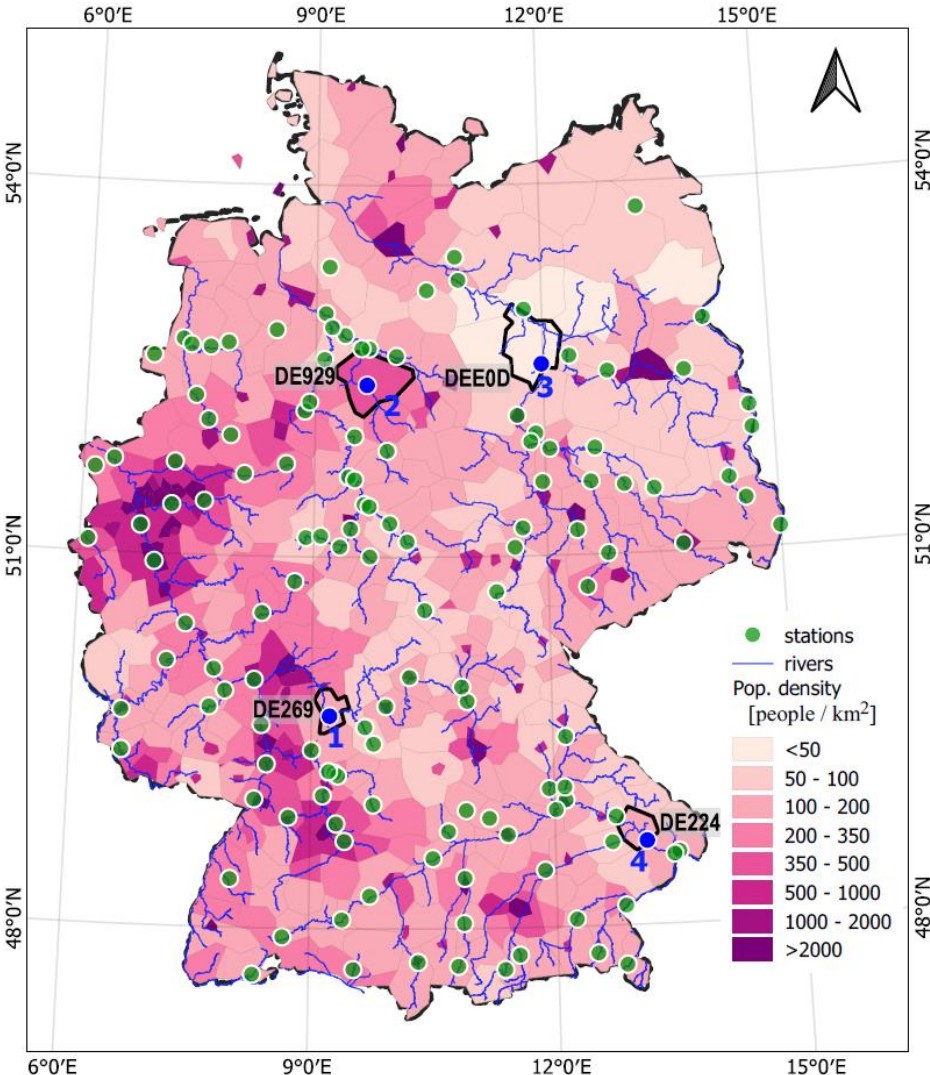

**Figure 1: Case study region Germany: Main rivers and tributaries, streamflow stations used and population density at the NUTS3 (Nomenclature of Territorial Units for Statistics 3) level. We selected 136 streamflow stations for which we could establish the relationship between flood peak and direct damage. Stations numbered in blue and contained**

**in their NUTS3 regions highlighted in black, are selected to provide a more detailed illustration of the forecasting results.**

## 3 Methods and Data

### 3.1 Concept and models used

140 To demonstrate how seasonal flood forecasting could be exploited by the insurance sector, we develop a model chain for our case study area Germany. The model chain builds on existing models and their output at the European scale and consists of the following workflow (Fig. 2): (1) The non-stationary flood frequency model of Steirou et al. (2022) provides flood peak distributions for 136 stations in Germany for the winter season as function of a climate indicator (autumn precipitation in this case). (2) The flood risk model of Alfieri et al. (2015) provides loss

145 for several return periods (10, 20, 50, 100, 200, 500 years) for areas associated with the 136 streamflow stations. Losses that belong to the same NUTS3 area are aggregated to calculate EAD (expected annual damage) at the NUTS3 level for Germany. (3) Both outputs are combined in a seasonal forecasting model that predicts, based on autumn precipitation, the deviation of EAD in the upcoming winter from the long-term average EAD. (4) The "Dynamic Integrated Flood Insurance" (DIFI) model of Hudson et al. (2019) for EU member states is used to

150 calculate the insurance premiums for the upcoming season as function of the forecasted EAD deviation at the NUTS3 level. (5) The flood risk model of Ward et al. (2017) provides the input data for DIFI. This workflow results in climate-informed insurance premiums, i.e. premiums that vary from year to year given a certain climate state.

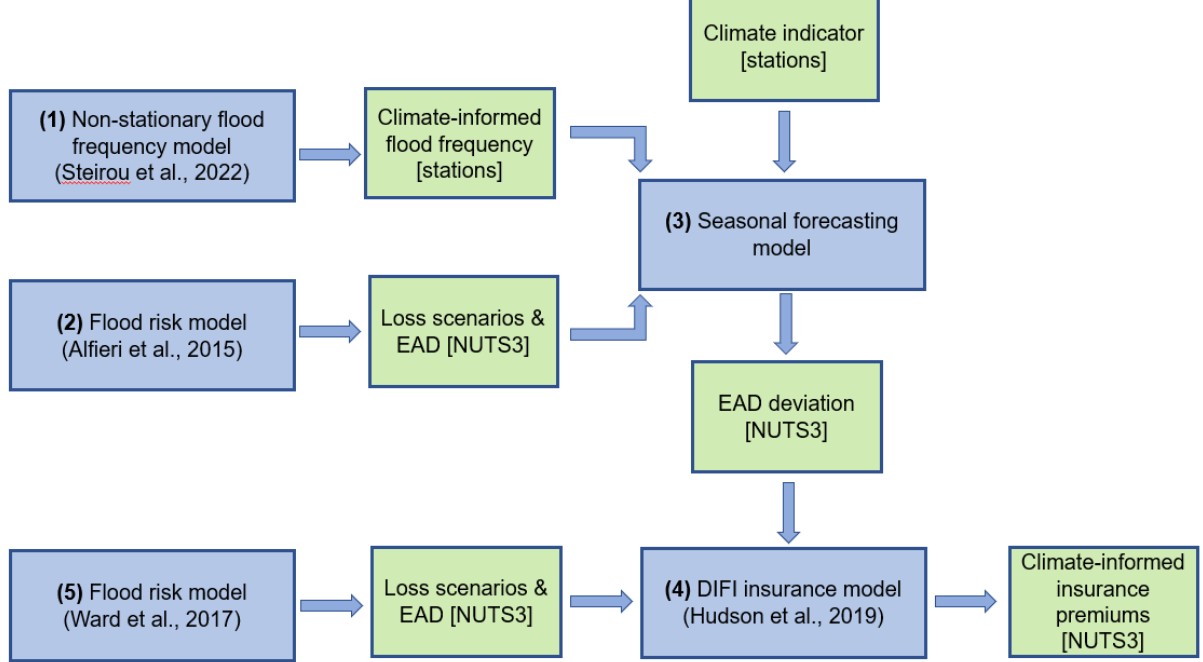

155 **Figure 2: Flow diagram of models used. The study builds on available models for frequency analysis, flood risk assessment and insurance for Europe.**

### 3.2 Seasonal forecasting of flood peaks and damage

We use the non-stationary flood frequency model developed by Steirou et al. (2022) to obtain climate-informed flood peak distributions for each streamflow gauge. They fitted the Generalised Extreme Value (GEV) distribution

160 to observed seasonal flood peaks by conditioning GEV parameters on the season-ahead climate covariate. They selected streamflow stations that covered the period 1950 to 2016 with at least 50 years of data. A set of 14 indices

was considered as potential season-ahead covariates including, for example, the North Atlantic Oscillation, the Sea Ice Concentration between 70 and 80° N, 30 and 105° E, or the precipitation. These covariates had either been shown to influence atmospheric circulation and related variables, particularly precipitation, over Europe or they affect flood generation directly as they represent the season-ahead catchment wetness. For several indices, these climate-informed models were preferred over the classical flood frequency approach with time-constant GEV parameters for many European regions. For instance, the climate-informed GEV with the location parameter conditioned on autumn precipitation was a better model for the winter flood peaks for 90% of the catchments in Northern Germany and the Netherlands.

In this paper, we use the results of Steirou et al. (2022) for the German streamflow stations. To illustrate the idea of seasonal forecasting of flood peaks and damage, we limit our study to one of the 14 indices of Steirou et al. (2022). Specifically, we estimate the probability distribution of flood peaks in winter (Dec–Feb) where the GEV location parameter is conditioned on the precipitation in the season-ahead (Sep–Nov). Thus, the winter flood peak related to a given return period varies as function of autumn precipitation; for instance, more precipitation in autumn increases the catchment wetness, thus increasing the peak of the 200-year flood (Jongman et al., 2014). The source of predictability of the seasonal forecasting model is thus the memory of the catchment which influences the antecedent conditions of flooding in the next season. The selected seasonal forecasting model is based on observed winter flood peaks at streamflow stations (Fig. 1) and observed nearby autumn precipitation. Flood peaks are derived from the GRDC discharge dataset. Autumn precipitation is obtained from the grid cell closest to the gauge location of the CRUTS4.02 dataset (Climatic Research Unit of the University of East Anglia).

The model of Steirou et al. (2022) is based on statistical relationships between climate indicators and the flood behaviour in the following season. In contrast, almost all national and international forecasting centres rely on dynamical models for seasonal climate forecasts, and much more effort and resources have been invested in developing dynamical forecasting systems in the last decades (Cohen et al., 2018). Data-driven approaches still have their justification, as they are much easier to implement and apply, and allow to efficiently search for states, regions or timescales associated with forecast skill (Cohen et al., 2018). As our study is intended as a proof of concept of how seasonal forecasting information could be used for the insurance sector, we follow an opportunistic approach and use an available and comparatively simple model.

In order to assess the potential benefits of climate-informed forecasting models for (re-)insurance, we extend the hazard model to a risk model across Germany, exploiting the earlier work of Nguyen et al. (2020). Specifically, we selected 136 stations in Germany for which we could establish the relationship between flood peak and direct damage. To establish this relationship, we use the results of 2D hydraulic simulations and damage evaluations by Alfieri et al. (2015). The 2D hydraulic simulations for specific return periods were based on the LISFLOOD-FP model. The direct economic damage for various sectors, such as residential, commerce, industry, transport, infrastructure, and agriculture, were estimated via country-specific depth-damage functions for different land use classes (Alfieri et al., 2015). To account for regional variations in asset values within each land use class, the depth-damage functions were adjusted using the Gross Domestic Product (GDP) Purchasing Power Standards of 2007. We assessed the damage at a resolution of 100 meters for selected return periods (T = 10, 20, 50, 100, 200, 500 years) and then aggregated the results to a 5 km resolution using the Areas of Influence method (Alfieri et al., 2015). The calculation of flood damage was performed for the area upstream of each river station. To consider the effects of flood protection measures, damage is set to zero for a given river section when the return period of the peak discharge is lower than its flood protection level. Flood protection levels were taken from Jongman et al.

(2014). Subsequently, we employ the relationship between flood flow (or its return period) and damage to compute the expected annual damage (EAD) for the selected stations and their respective areas.

In this way, we consider two cases: the climate-informed case where the flood probability distribution, and consequentially EAD, change from year to year in response to changes in autumn precipitation; and the traditional approach or unconditional case where flood probability distribution and EAD are constant in time. By aggregating the estimated station-based EAD values, we calculate EAD at the NUTS3 level for Germany for both cases. We then introduce the forecasted (percentage) deviation in EAD (FDE) defined as the ratio between the climate-in-

formed EAD and the unconditional EAD for each of the 401 NUTS3 units in Germany (Fig. 2). Thus, FDE is a time-varying metric, reflecting changes in the flood probability over time due to interannual changes or long-term trends in climate.

### 3.3 Insurance model (DIFI)

In order to translate the forecasted deviations in EAD into forecasted deviations in insurance premiums, we use

the "Dynamic Integrated Flood Insurance" (DIFI) model (Hudson et al., 2019). The DIFI model has been developed for the EU member states and combines flood risk information with an insurance sector and household behaviour model. The model simulates premiums at NUTS3 resolution, and has been recently applied to assess the effect of climate change on unaffordability of premiums and insurance uptake under various existing and proposed flood insurance market structures in Europe (Tesselaar et al., 2020). For an extended description of the model, we

refer to Hudson et al. (2019) and Tesselaar et al. (2020); here we briefly summarize the main components.

DIFI considers the flood insurance scheme in Germany. Flood insurance in Germany is provided by private insurers, who charge risk-based premiums, and obtaining insurance coverage is optional for households. In this system, private insurers compete to charge accurate risk-based premiums in order to increase their market share, while limiting risk of adverse selection. Private insurers calculate premiums based on the local EAD and the uncertainty of this risk. Concerning the latter, insurers generally apply a surcharge based on the variance of flood risk to cover

their risk aversion. To combat moral hazard, insurers generally apply a deductible of 15% of flood damages (Paudel et al., 2013). To reduce the risk of extreme events, insurers generally cover this risk on the re-insurance market in a stop-loss mechanism, where re-insurers cover losses after a certain threshold of losses have occurred in a given time frame. Following Paudel et al. (2013), for private flood insurance systems in Europe, on average, this thresh-

old is set at 85% of annual losses. This means that 15% of annual losses are transferred to re-insurance firms against an annual premium. The calculation of re-insurance premiums is similar to general premiums, except that re-insurers may charge a profit loading factor due to their considerable market power.

The calculation of insurance premiums at the NUTS3 level is based on EAD, which is obtained from the flood risk model GLOFRIS (Ward et al., 2017; Winsemius et al., 2013). The GLOFRIS model applies an established ap-

proach to simulating flood risk, by combining flood hazard, exposure, and vulnerability. To simulate the seasonal fluctuation of flood insurance premiums,, we apply the forecasted percentage deviation of EAD assessed in this study to the baseline EAD (for the year 2010) obtained from GLOFRIS. Moreover, an important parameter in the estimation of flood insurance premiums is the uncertainty of risk estimates, which is captured by the standard deviation of EAD estimations, obtained using the approach explained in Hudson et al. (2019). Note that in our

assessment, premiums are set for households located in 100-year floodplains, aggregated to the NUTS3 level. This means that premiums are reflective of the average risk of all such floodplains within a NUTS3 region. To estimate premiums on this level, we divide the NUTS3 region's EAD and its variance by the number of households that are

exposed to the 100-year flood. The number of exposed households is re-scaled for each forecasting year within the period considered (1950 to 2016) using data from Paprotny & Mengel (2022).

## 4 Results and Discussion

### 4.1 Forecasting flood probability distributions and EAD

Our model setup provides annually varying flood probability distributions and associated EAD values for all selected streamflow gauges in Germany. Fig. 3 exemplarily shows these results for four gauges selected from the major river basins Rhine, Weser, Elbe and Danube. To visualise the variation of the flood probability distribution, we show the 100-year flood discharge (HQ-100) as an example. Both variables (HQ-100 and EAD) vary substantially in time indicating an important role of the season-ahead climate state on flood probabilities and risk. The HQ-100 of the climate-informed model varies along with the covariate (autumn rain) as the flood probability distribution in winter is conditioned on the preceding autumn precipitation. It thus fluctuates around the time-constant HQ-100 discharge of the unconditional model which remains unaffected by the climate covariate. At three gauges (ID=1,2,4), the mean HQ-100 of the climate-informed approach is similar to that of its unconditional counterpart. However, at gauge ID=3, the HQ-100 of the unconditional case is noticeably larger.

Comparing the forecasted HQ-100 (second row in Fig. 3) with the observed flood peaks (third row in Fig. 3), a modest association is seen. This is explained by two effects. Firstly, we compare a probability distribution, and more specifically one return level value (HQ-100), against observed data. The observations represent a single realisation of the true, but unknown distribution, and another realisation would look quite differently. Secondly, temporal changes in winter flood probabilities are not only influenced by the selected covariate (autumn precipitation), but are also affected by other variabilities, such as precipitation and snow accumulation and melt in winter. Fig. 3 (first, third, and fourth rows) reveals a close and positive association between the season-ahead autumn precipitation and the forecasted HQ-100 and EAD values. Higher catchment wetness in autumn is transferred to higher HQ-100 values, which are translated in turn into higher damages and thus higher EAD values. Comparing the variability of the autumn precipitation, the HQ-100 and EAD suggests that the temporal variability increases along this process chain. Relatively small variations in autumn precipitation are transferred into slightly larger variations in HQ-100, but substantial variations in EAD.

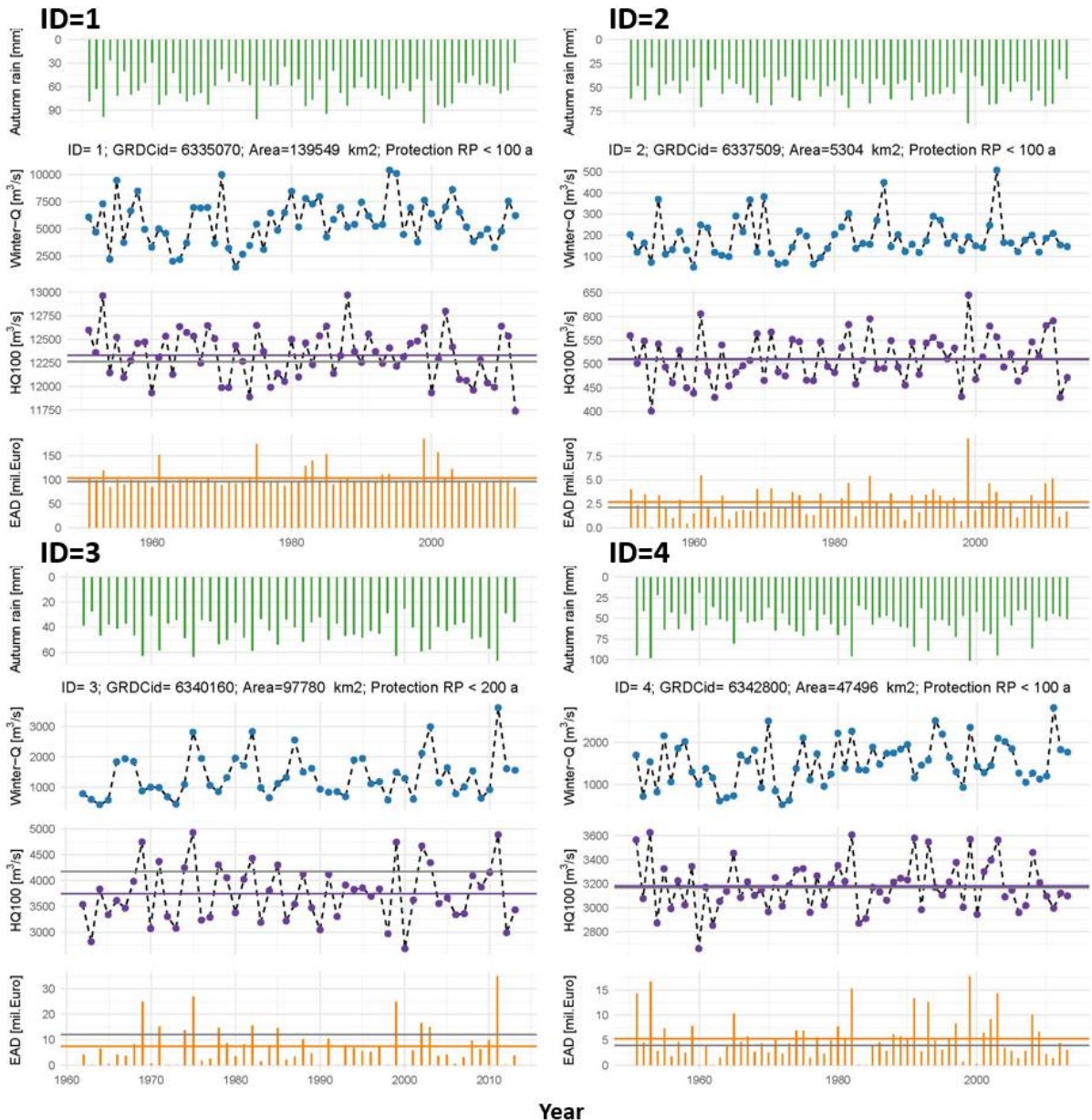

270

**Figure 3: Forecasting results – example time series at four gauges (location given in blue in the maps of Fig. 2 and 4). Shown is autumn rain (first row), winter peak flow (second row), forecasted 100-year flood discharge (HQ-100, third row) and forecasted EAD (fourth row). In the third and fourth rows the average climate-informed HQ-100 and EAD values are shown in purple and orange horizontal lines while the unconditional HQ-100 and EAD values are shown in**
275 **grey. Information on catchment size and protection level is also provided.**

The two approaches, unconditional and climate-informed, can lead to significant differences in the long-term EAD (e.g. the latter is reduced by 37.5% for ID=3). This effect is explained by the non-linearity of the relation between flood peaks and damages (ID=2,4) and, in addition, by the difference in the average flood quantiles estimated by the two approaches (ID=3).

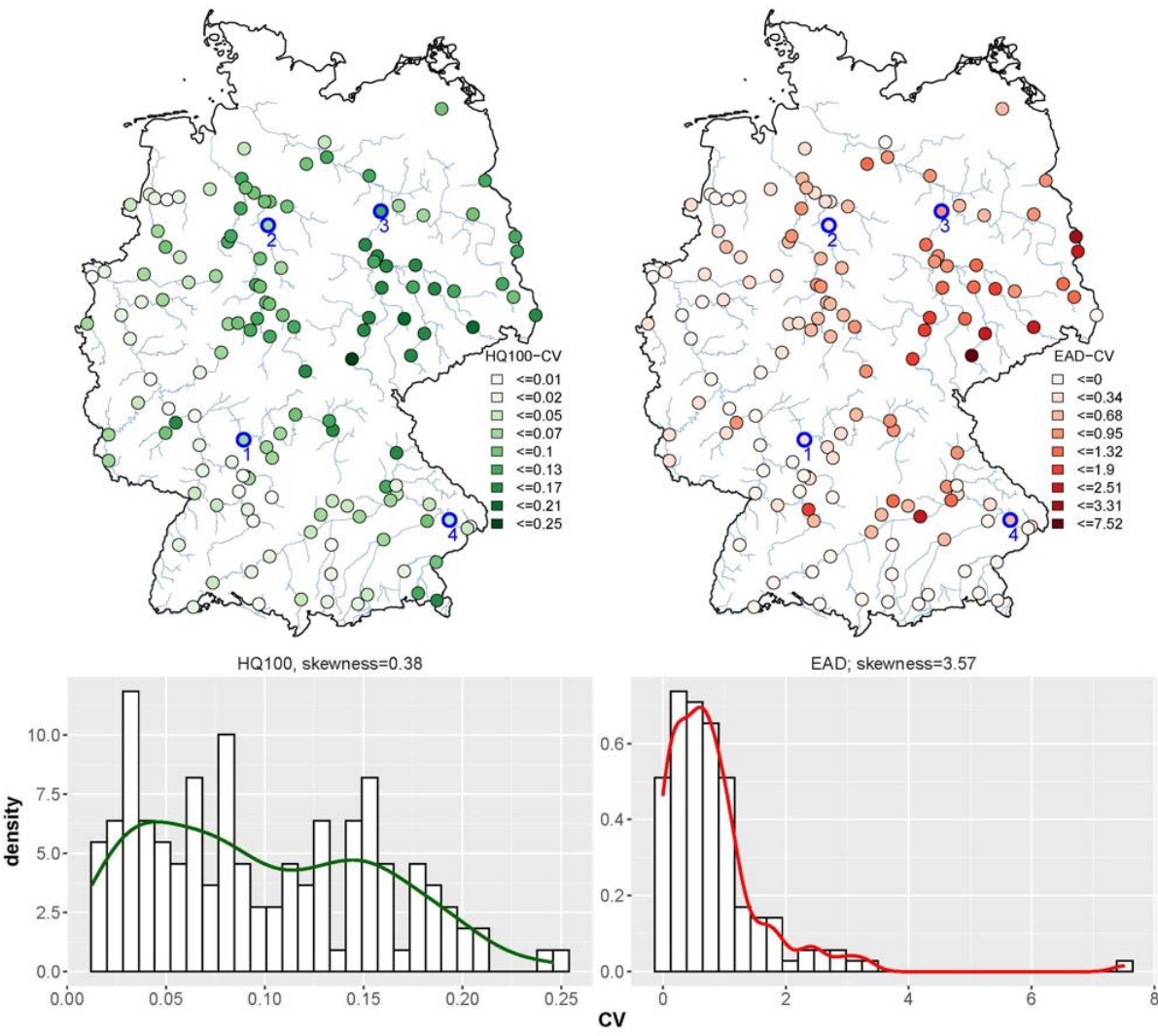

**Figure 4: Coefficient of variation of the 100-year flood discharge HQ-100 (left) and expected annual damage EAD (right) for 136 stations. Maps in the top row display their class breaks estimated using Jen's Natural Breaks optimization. Corresponding histograms and density curves are shown in the second row.**

To understand the temporal variation of the flood quantiles and the damage for the entire set of stations, Fig. 4 shows the coefficient of variation (CV) for the forecasted HQ-100 flood discharge and EAD of the climate-informed model across Germany. For HQ100, higher variation is found in the Elbe, Oder and Weser basins and the downstream areas of the Danube compared to the Rhine and upstream areas of the Danube river basins. The variability of the forecasted HQ100 discharge results from two sources: the variability of the autumn precipitation and the variability of the observed flood peaks. Comparing both (Fig. A1) reveals that the variability is dominated by the latter source: The spatial pattern of CV for the forecasted HQ100 is very similar to the pattern of CV of observed flood peaks. The latter, in turn, depends on a range of factors, such as catchment size, climatic regime, flood generation processes and influence of reservoirs and flood retention basins (Rosbjerg et al., 2013).

The spatial pattern of the variability of EAD follows the pattern of HQ-100 variability. To measure the association between the two maps, Kendall's tau is used and indicates a strong correlation of 0.59 between the HQ-100-CV and EAD-CV values. There is a significant relationship (p-value < 0.001) between the variation in HQ-100 flood and the variation in EAD across the different gauges in Germany. However, the variability in EAD is one order of magnitude larger than the variability in HQ-100. The CV values of HQ-100 display a weak positive skewness (0.38) with values up to 0.25. On the other hand, the CV of EAD exhibits a strong positive skewness (3.57),

indicating a highly skewed distribution with a longer tail on the right side. This increase in variability can be explained by the non-linearity of the relation between flood peaks and damages. Whenever flood protection levels are exceeded, damage jumps from zero to large values. Shifting the location parameter of the flood probability distribution thus tends to lead to substantial changes in EAD.

**4.2 Forecasting insurance and premiums**

Figure 5 (left) presents the spatial variability of household insurance premiums across Germany, based on the flood risk – insurance model (Fig. 2). Most premiums are about 1200 Euro per household with a few exceptions of up to 5000 Euro. In some NUTS3 regions flood risk, and therefore premiums, is 0. This is explained by the fact that the flood risk model GLOFRIS only considers the flooding of large rivers (with a Strahler order of 6 and higher). There are some NUTS3 regions without large rivers. From an insurance perspective, this is not necessarily a limitation of the model, since insurance for flooding of rivers is often a separate product from flood insurance for flash-floods and pluvial floods. Because we use the flood risk per capita to calculate premiums, regions with relatively high total risk do not necessarily show high premiums: for high-risk areas with high population density premiums are low. The opposite is of course equally possible. An important note is that premiums are risk-reflective, meaning that exposed households pay for the risk they face which may be higher than in reality due to possible subsidizing effects lowering premiums in some areas. However, because the analysis is done on the NUTS3 level, the degree to which premiums reflect actual risk is not highly accurate. For example, for the households that are exposed to a 100-year flood or worse there is a degree of risk sharing, whereas households that face little inundation associated with this flood pay relatively high premiums compared to households that face worse levels of inundation. On the other hand, the level to which premiums are risk-reflective in reality is often not very detailed either. Figure 5 (right) shows the volatility of projected premiums, reflecting the variation in forecasted annual flood risk. This map shows which regions are prone to larger variation in forecasted flood peaks and damages than other regions and how this plays out in terms of the variation in forecasted premiums. This information is highly relevant for insurers to prepare for the upcoming year and set premiums such that they already anticipate expected risk for the next year. Furthermore, these forecasted variations in premiums may also support (re-) insurers in their strategy to reserve capital for upcoming events and losses. It should be noted that our forecast only considers changes in future flood peaks based on autumn precipitation and does not include other factors that may increase or decrease forecasted risk and premiums.

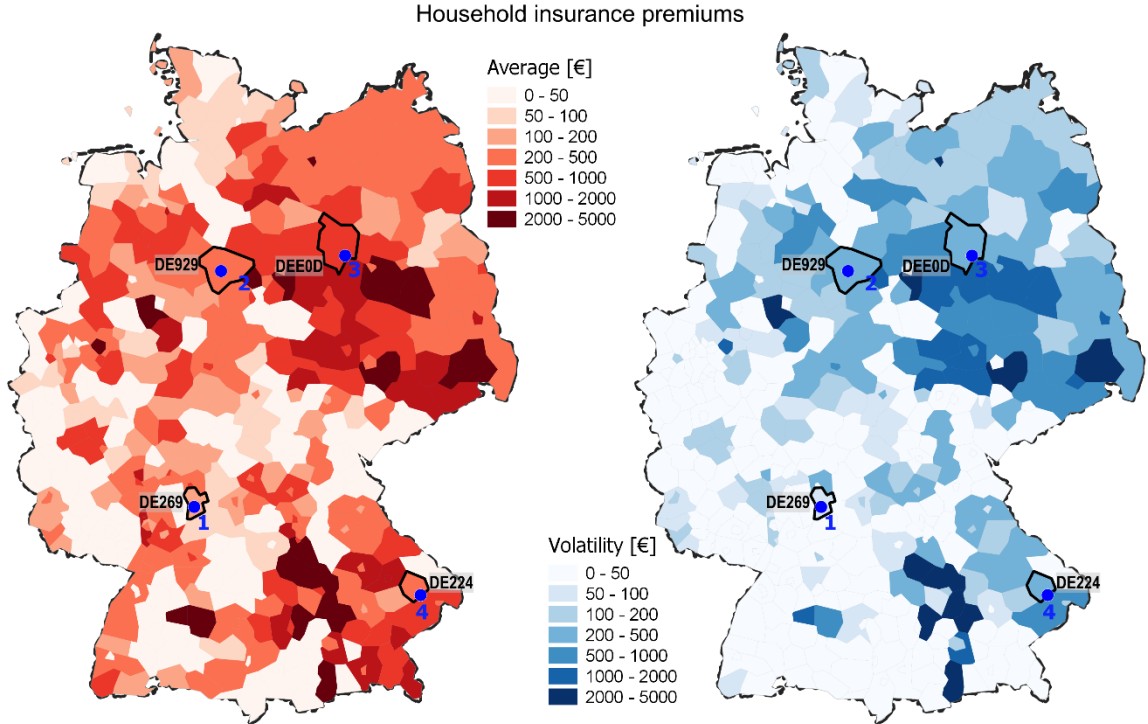

**Figure 5: Average household insurance premiums (left) and volatility (i.e. standard deviation) of premiums (right) at the NUTS3-level.**

Figure 6 illustrates how premiums, based on the novel seasonal forecast developed in this study, may deviate from premiums calculated based on a rolling average of (in this example) 10 years in the four NUTS3 regions: Miltenberg (NUTS3 ID: DE269), Hannover (DE929), Stendal (DEE0D), and Deggendorf (DE224). These four regions contain the four streamflow gauges shown in Figure 3. If premium-setting applies a moving average method, as recommended for the insurance of climate risks (Lyubchich & Gel, 2016), the premium changes annually, as it is considerably influenced by amounts of damage in recent years. However, premiums change within a comparably small range (Figure 6). If, instead, premiums are based on seasonal forecasts, positive and negative changes are much larger. Insurers may charge higher premiums in years with expected high risk, causing policy-holders to transfer to insurers that do not accurately forecast risk, but follow the rolling average approach. Vice versa, in years of low expected flood risk, premiums may be adjusted accordingly, and the insurer may attract policy-holders back from insurers that maintain a rolling average to calculate the premium. Viewed in this way, the accuracy of risk forecasting improves the competitive position of insurers.

From a societal perspective, a premium that accurately reflects expected risk may trigger adaptive behavior of policyholders. A sudden increase in the flood insurance premium communicates to policy-holders the potential looming threat. An effective measure to encourage adaptation in years of severe expected risk is to allow policy-holders to reduce their premium by applying certain risk-reduction measures. Such measures may include applying flood-resistant building materials, relocating vulnerable possessions to higher floors, or installing flood barriers that prevent flood water from entering the building. By stimulating risk-reduction effort by policyholders, insurers may reduce the actual spike in flood risk in years when risk is expected to be high, while at the same time limiting the costs for policyholders. Therefore, if implemented correctly, premiums that reflect accurately forecasted flood risk may increase resilience and reduce overall costs of flood risk.

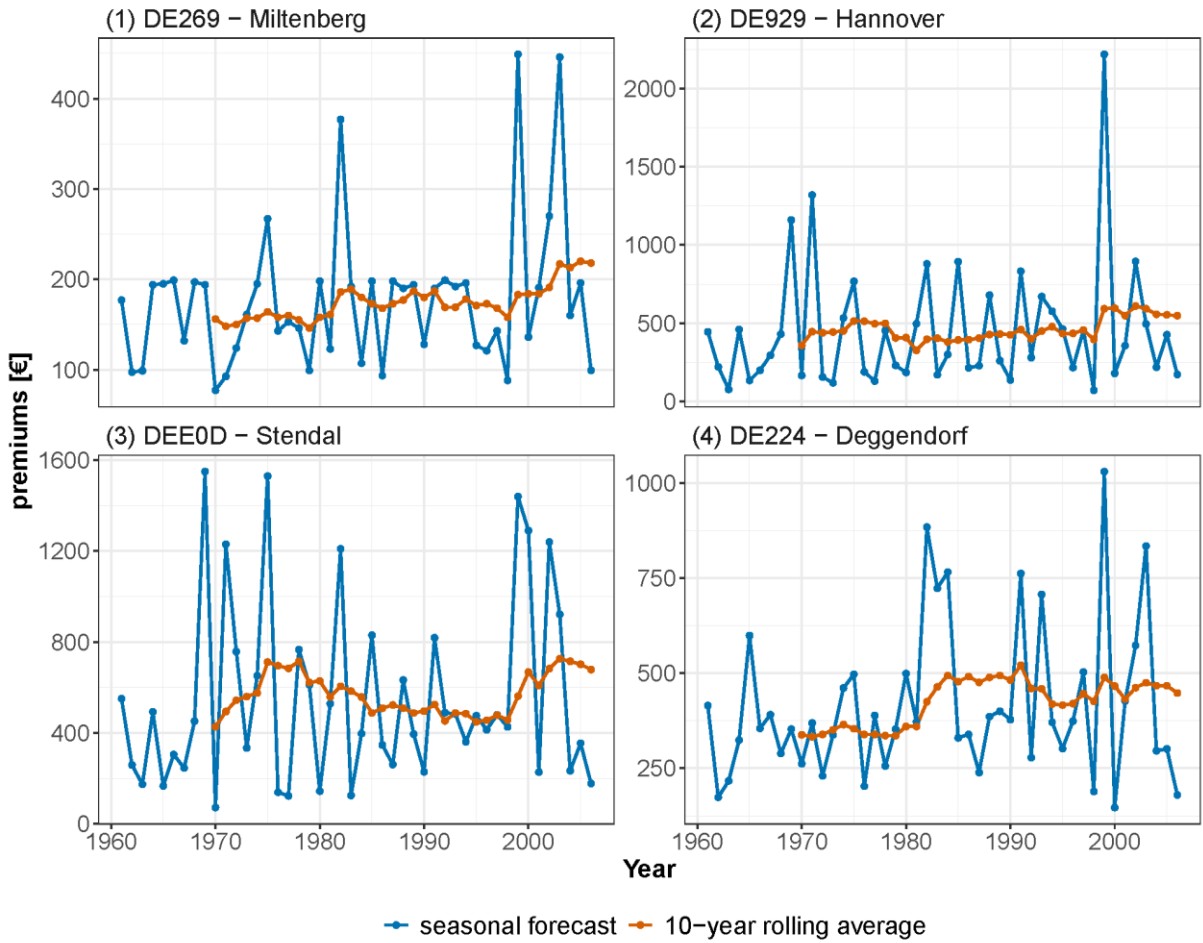

**Figure 6: Time series of premiums for the four illustrative NUTS3 regions (locations are shown in Fig. 1 and Fig. 5).**

### 4.3 Limitations and recommendations

The potential of seasonal flood forecasts for the insurance industry has hardly been investigated so far. Here, we address this knowledge gap by exploring the sensitivity of household premiums to temporal changes in flood probabilities. Our study suggests that these variations can be substantial, leading to significant changes in flood risk and premiums. Hence, these results indicate a potentially interesting lever for the (re-)insurance sector worthwhile of further investigation.

Our explorative study has a number of limitations, some of which follow from its use of available models and datasets. Firstly, we apply models that have been developed, calibrated and validated by earlier studies for different purposes. We combine these models, or their outputs, without performing an in-depth calibration and validation of our model chain. Our study is intended as proof of concept and not as decision tool for the insurance sector. In the latter case, one would need to perform a rigorous model evaluation including sensitivity and uncertainty analysis to understand the skill of the risk forecasts. For example, the aim of Steirou et al. (2022) was to investigate whether the seasonal flood peak distribution was significantly influenced by the catchment or climate state of the season ahead. Hence, they focussed on whether the climate-informed distribution was a better fit than the traditional, stationary distribution considering model complexity, but they did not perform an in-depth analysis of model performance. Fig. A2 shows Q-Q-plots for the climate-informed flood frequency distribution for the four example streamflow stations (ID=1,2,3,4). Although these examples suggest that the model of Steirou et al. (2022) agrees well with observations, the application of our model chain would require more elaborate model assessments.

Secondly, the climate-informed flood frequency model by Steirou et al. (2022) could be improved. While it considers a single climate covariate only, the joint use of two covariates, one representing climate state and one representing catchment state in the season ahead, might improve the forecasting skill. Moreover, the climate covariate, autumn precipitation, is extracted from a gridded dataset (CRU TS4.02 data; Harris et al., 2020). Using catchment-averaged precipitation could be a more representative proxy for the catchment wetness. Furthermore, instead of a non-stationary extreme value distribution approach, a different method could be used to forecast the flood peak probability distribution. The forecasting method could be based on dynamical models or AI-based approaches (Cohen et al., 2022).

Thirdly, the DIFI insurance model currently uses EAD data from the GLOFRIS flood risk model of Ward et al. (2017). It then applies a yearly deviation of the EAD data based on our forecasting model, which is built on the flood risk model of Alfieri et al. (2015). Although both flood risk models are conceptually similar, future research can further improve consistency across the different (sub-) models and data by utilizing the same data for forecasting simulations.

Fourthly, our model assumes that vulnerability is constant in time and that climate is the only driver of changes in flood hazard. It thus ignores other changes in time, such as improved flood protection and early warning systems. While such time-varying effects could be included in principle, their addition would substantially increase the uncertainty. Furthermore, we limit our forecast model to the next season only while premiums are calculated for the next year. Finally, our case study area Germany is not the best location in terms of seasonal streamflow forecasting skill. The proposed approach seems to be of even more practical use in areas outside of Europe, as other regions show more pronounced links between climate variabilities and flood characteristics (Arnal et al., 2019, Yan et al., 2020). While these limitations may reduce the value of the specific quantitative results for our case study, they do not affect the general insights gained in our study. Although our results depend on the characteristics of Germany in terms of climate, hydrology, flood protection, flood-prone assets, damage processes and insurance system, the concept and methodology are transferable to other regions.

In the pricing of insurance premiums, (perceived) uncertainty in flood risk assessment estimates generally leads to higher premium settings by insurers (Kunreuther et al., 1993). We identify two main sources of uncertainty in our seasonal forecasting concept that are relevant for pricing: (1) Our results indicate that some areas show higher variations in historical forecasted EADs than others. These variations may reflect natural variations in the hydro-climate system or human interventions in catchments and rivers. For instance, the operation of reservoirs may decrease the temporal variation of flood flows by cutting flood peaks. Having a better understanding of these variations and their geographic location can be a basis for premium differentiation where the areas with larger fluctuations in historical forecasted EADs have somewhat higher premiums surcharges. (2) Another source of uncertainty not yet addressed in our concept is modelling uncertainty. This may be captured through the employment of probabilistic forecasting distributions. However, tailoring this information into risk assessment and pricing is challenging and needs careful processing and communication (e.g. Hansen et al., 2022).

Another recommendation for future research is to run the presented modelling cascade with and without information about the uncertainty in forecasted EAD. For example, such comparative runs would provide insights in whether prior information on EAD leads to lower (higher-) premiums in some locations directly affecting policy holders. Furthermore, from the perspective of the insurers, it may be interesting to assess how higher (lower-) forecasted premiums lead to higher (lower-) capital reserves to anticipate forecasted risks (Maynard, 2016).

## 5 Conclusions

This study addresses a topic which has not received much attention, namely to which extent seasonal flood fore-
casts could be used to inform insurance schemes. In this explorative study, we demonstrate in an application for
Germany how seasonal forecasting of flood peaks and flood risk (expected annual damage EAD) influence insur-
ance premiums. Our results indicate that interannual variations of flood peaks and risks are substantial, leading to
substantial variations in household insurance premiums. This variation provides a potential lever to inform insurers
and policy holders about upcoming changes in flood risk and to device strategies to hedge against such variations.
Most premiums are about 1200 Euro per household with a few exceptions of up to 5000 Euro. Differences can be
explained by a combination of both the variations in forecasted EAD and the number of exposed households living
in the region. Most fluctuations in predicted premiums are found in the eastern part of Germany. Better under-
standing of these historical variations in forecasted EAD and their geographic distribution can be a basis for pre-
mium differentiation and to optimize premium settings. In order to further explore the potential of seasonal flood
peaks and EAD forecasts, future research can tap into probabilistic forecasting distributions and use these as ad-
ditional information for pricing risk. However, tailoring this information into risk assessment and pricing is chal-
lenging and needs careful processing and communication in order to apply the method in operational risk assess-
ment and pricing.

**Appendix**

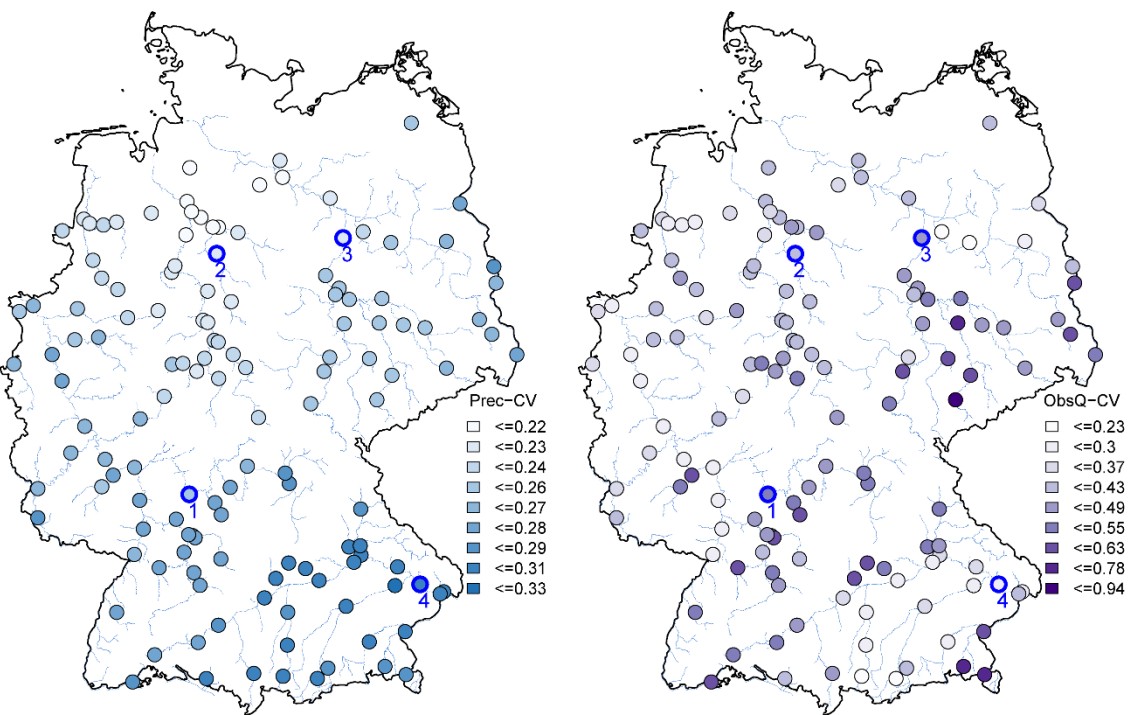

**Figure A1: Coefficient of variation of the autumn precipitation (left) and observed flood peaks (right) for 136 stations. Maps display their class breaks estimated using Jen's Natural Breaks optimization.**

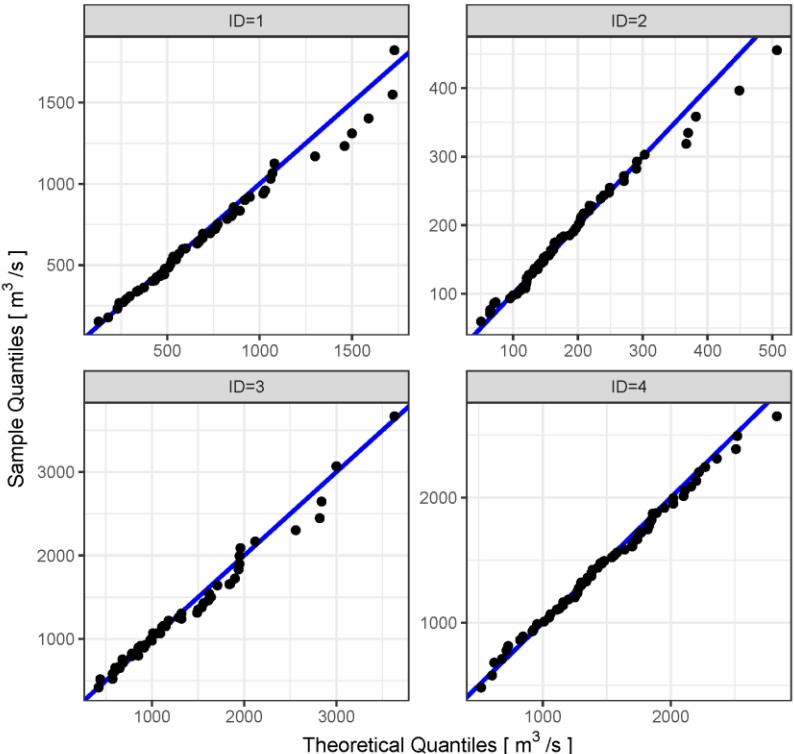


**Figure A2: Q-Q plots for the four gauges shown in Fig. 3. These plots demonstrate that the model provides a good representation of the observations, although some overestimation can be seen for gauges ID=1 and ID=2.**

**Data avaibility**

The GRDC discharge dataset was obtained from the Global Runoff Data Centre, 56068 Koblenz, Germany (https://www.bafg.de/GRDC/EN, last accessed: October 2017) and was recently made available for online download via https://portal.grdc.bafg.de. Flood hazard maps for the European Union can be downloaded from https://data.jrc.ec.europa.eu/collection/floods. Flood protection levels are taken from Jongman et al. (2014). Grid-
based precipitation data were extracted from the CRUTS4.02 dataset from the Climatic Research Unit (CRU, https://crudata.uea.ac.uk/cru/data/hrg/, last accessed: April 2019) of the University of East Anglia. The exposure dataset is available through Paprotny et al. (2022).

**Author contributions.**

BM, DN and JA conceptualized the study. DN performed the analysis on flood forecasting and expected damage. MT calculated the premiums. DN prepared the manuscript with contributions from all co-authors. All authors have read and agreed to the published version of the manuscript.

**Competing interests.**

At least one of the (co-)authors is a member of the editorial board of Natural Hazards and Earth System Sciences.

**Acknowledgements**

We acknowledge the support of the Humboldt Foundation and the projects ClimXtreme (Module C Impacts – Subproject 5 FLOOD, BMBF, 01LP1903E and 01LP2324E), AXA Research Fund "The link of flood frequency to catchment state and climate variations" – joint research initiative between AXA Global P&C and GFZ, Potsdam and ERC advanced grant COASTMOVE (nr 884442).

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
