# Peer review of "Exploring the use of seasonal forecasts to adapt flood insurance premiums"

_EGUsphere, 2023_

## Author Comment (AC1)

*Response to Reviewer #1*

*(Comments by Reviewer #1: https://doi.org/10.5194/egusphere-2023-2862-RC1)*

We would like to thank Reviewer #1 for the thoughtful and constructive feedback on our manuscript entitled "Exploring the use of seasonal forecasts to adapt flood insurance premiums" (*https://doi.org/10.5194/egusphere-2023-2862*). Please find our point-by-point response below.

**General comments:**

The manuscript "Exploring the use of seasonal forecasts to adapt flood insurance premiums" by Viet Dung Nguyen et al. represents a significant and innovative contribution to the field of flood insurance by integrating seasonal flood forecasts into the adjustment of flood insurance premiums. This approach is both timely and highly relevant in the context of a potential increase in flood risks associated with climate change.

The manuscript stands out for its novel integration of hydrological and economic models. The use of non-stationary flood frequency analysis coupled with the Dynamic Integrated Flood Insurance (DIFI) model showcases an interdisciplinary approach that is important for addressing challenges in this kind of climate risk management. Additionally, the study's focus on Germany provides a valuable case study that offers insights into the application of these models in a specific geographic and climatic context.

The paper is well-structured, progressing logically from a detailed introduction that sets the stage for the research, through to the methodology, results, and a comprehensive discussion. The authors also discuss the limitations and potential implications of their findings, which is crucial for a balanced scientific discourse.

**Response:**

*We appreciate the reviewer's positive overall assessment of our work.*

**Specific comments (RC)**

**Comment #1 on "Seasonal forecast choice"**

The authors explain that there are two approaches for seasonal flood forecasting, one based on dynamical models and another one based on statistical approaches. The authors note in line 70, "seasonal climate forecast from dynamical or statistical models have seen great advancements in recent years," and again in line 89, "Given the progress in recent years in seasonal flood forecasting methods…" However, the manuscript stops short of detailing these advancements or providing specific examples and references to substantiate these claims.

- I'd suggest that the authors further develop these claims (with specific examples and references).
- Linked to the previous point, I'd also suggest including some lines or a paragraph justifying the authors' choice of going for a statistical approach with past data instead of directly using data from dynamical seasonal models. Factors might involve the comparative skill of statistical models against dynamical systems in the study area, the

availability and resolution of historical data, limitations inherent to dynamical models that make them less suited to the specific objectives of this research, the availability of hydrological models or the scope of the paper as a proof of concept.

**Response #1**

*In the revised manuscript, we have extended the text and provide relevant references to support our statements as follows:*

*"… Against this background it is important to note that seasonal climate forecasts from dynamical or data-driven (statistical and AI-based) models have seen great advancements in recent years. For dynamical forecasting models, improvements in predictive skill have mainly benefited from improved physical process representation and model initialization, the emergence of ensemble forecasts representing uncertainty and computing advances (Jia et al., 2015, Bauer et al., 2016, Zhang et al., 2023). Data-driven seasonal forecasting has benefited from the improved estimation of initial hydrologic conditions and incorporation of climate information, as well as the advent of large datasets and AI-based forecasting algorithms capable of handling nonlinear relationships (Mendoza et al., 2017, Huang et al., 2020). Meanwhile, the majority of dams in the United States, for example, rely on seasonal forecasts of reservoir inflows to decide about water release (Turner et al., 2020)…"*

*Bauer, P., Thorpe, A. & Brunet, G. The quiet revolution of numerical weather prediction. Nature 525, 47–55 (2015). https://doi.org/10.1038/nature14956*

*Huang, Z., Zhao, T., Liu, Y., Zhang, Y., Jiang, T., Lin, K., & Chen, X. (2020). Differing roles of base and fast flow in ensemble seasonal streamflow forecasting: An experimental investigation. Journal of Hydrology, 591, 125272.*

*Jia, L., and Coauthors, 2015: Improved seasonal prediction of temperature and precipitation over land in a high-resolution GFDL climate model. J. Climate, 28, 2044–2062, https://doi.org/10.1175/JCLI-D-14-00112.1*

*Mendoza, P. A., Wood, A. W., Clark, E., Rothwell, E., Clark, M. P., Nijssen, B., Brekke, L. D., and Arnold, J. R.: An intercomparison of approaches for improving operational seasonal streamflow forecasts, Hydrol. Earth Syst. Sci., 21, 3915–3935, https://doi.org/10.5194/hess-21-3915-2017, 2017*

*Turner, S. W. D., Xu, W., and Voisin, N.: Inferred inflow forecast horizons guiding reservoir release decisions across the United States, Hydrol. Earth Syst. Sci., 24, 1275–1291, https://doi.org/10.5194/hess-24-1275-2020, 2020.*

*Zhang, J., Guan, K., Fu, R., Peng, B., Zhao, S., & Zhuang, Y. (2023). Evaluating seasonal climate forecasts from dynamical models over South America. Journal of Hydrometeorology, 24(4), 801–814. https://doi.org/10.1175/JHM-D-22-0156.1*

*Additionally, we include the following text in section 3.2 to justify our choice of a statistical approach over dynamical models:*

*"… The model of Steirou et al. (2022) is based on statistical relationships between climate indicators and the flood behaviour in the following season. In contrast, almost all national and international forecasting centres rely on dynamical models for seasonal climate forecasts, and much more effort and resources have been invested in developing dynamical forecasting systems (Cohen et al., 2018). Data-driven approaches still have their justification, as they are*

*much easier to implement and apply, and allow to efficiently search for states, regions or timescales associated with forecast skill (Cohen et al., 2018). As our study is intended as a proof of concept of how seasonal forecasting information could be used for the insurance sector, we follow an opportunistic approach and use an available, comparatively simple model…”*

*Cohen, J., Coumou, D., Hwang, J., Mackey, L., Orenstein, P., Totz, S., Tziperman, E. (2019). S2S reboot: An argument for greater inclusion of machine learning in subseasonal to seasonal forecasts. WIREs Climate Change, 10(2), e00567. doi:https://doi.org/10.1002/wcc.567*

**Comment #2 on "Model selection"**

The choice of the non-stationary flood frequency model and the DIFI model is central to your study. It would be beneficial to provide a more detailed justification for selecting these specific models over other available models.

- How do these models compare in terms of their predictive accuracy, computational efficiency, and sensitivity to different climatic and hydrological conditions?
- Regarding the GEV distribution used from Steirou et al. 2022, have you checked how good does it fit to the 136 stations data? In Steirou et al. 2022 it seems that what is checked is the degree of improvement by using climate information in the location parameter in the GEV adjustment. But there is an inherent uncertainty on which is the best distribution fit when adjusting for return periods (a regular problem in hydrology, climate, and meteorology fields).

I'd suggest including some mention to this uncertainty. No need to perform additional computations, but Q-Q plots representing the observed quantiles vs. the theoretical quantiles from several distributions are a good tool to visualize this complexity.

**Response #2**

*Yes, Steirou et al. (2022) have focused on the improvement of using climate information and have not extensively discussed model selection and model validation. Although we agree that a more in-depth discussion of these topics would have been worthwhile in the paper by Steirou et al. (2022), we are reluctant to provide such an analysis and discussion in our study, because our study focusses on illustrating the concept of how seasonal flood forecast could be exploited by the insurance sector. Hence, we are not so much concerned with the skill of the different model components that we use in our application, but rather aim to demonstrate the possibilities for the insurance sector.*

*Below we provide Q-Q plots for the four gauges that are used to exemplarily demonstrate the results (in Figure 3). These plots demonstrate that the model provides a good representation of the observations, although with some overestimation for gauges ID=1 and ID=2. We prefer not to include these plots (and additional plots and discussion) in the manuscript in order not to distract from the conceptual framework of our paper. However, we include the figure to the appendix and add the following text to sub-section '4.3 Limitations and recommendations' to make sure that readers don't get a wrong understanding of our model chain:*

[Figure]

*Figure A2: Q-Q plots for the four gauges shown in Fig. 3. These plots demonstrate that the model provides a good representation of the observations, although some overestimation can be seen for gauges ID=1 and ID=2.*

*"... For example, the aim of Steirou et al. (2022) was to investigate whether the seasonal flood peak distribution was significantly influenced by the catchment or climate state of the season ahead. Hence, they focused on whether the climate-informed distribution was a better fit than the traditional, stationary distribution considering model complexity, but they did not perform an in-depth analysis of model performance. Fig. A2 shows Q-Q-plots for the climate-informed flood frequency distribution for the four example streamflow stations (ID=1,2,3,4). Although these examples suggest that the model of Steirou et al. (2022) agrees well with observations, the application of our model chain would require more elaborate model assessments..."*

**Comment #3 on "Methods and data section"**

This section, although extensive, currently falls short in providing a comprehensive and detailed description of the datasets and models employed. Data such the one coming from the Climatic Research Unit (CRU) and the Global Runoff Data Centre (GRDC), receive late mentions (line 339 for CRU and line 380 for GRDC) without prior introduction or elaboration on their characteristics. Similarly, models like GLOFRIS are introduced abruptly (line 278), leaving a gap in the reader's understanding of the methodological underpinnings of the study. I'd suggest reviewing this section and try to complete the missing information by, for example, creating a specific sub-section for 'Data and Models', leaving the 'Methodology' in another sub-section.

**Response #3**

*We agree with your observation regarding the need for a more comprehensive and detailed description of the datasets and models employed. In the revised manuscript, we restructured the paper to better balance the section lengths, and place 'Case study area Germany' in a separate section (new section 2). In addition, we introduce the data, such as CRU and GRDC, earlier, i.e. together with the introduction of the model component where they are used (sub-section '3.2 Seasonal forecasting of flood peaks and damage').*

*Additionally, the estimation of EAD using the GLOFRIS model is described in section 3.3. Although this approach is important to assess flood insurance premiums, it is not fundamental to the theoretical framework and novelty of this study. Therefore, we limit the description of GLOFRIS to mentioning its function in this study, but refer to the original publications for more information on the model.*

**Comment #4 on "Model calibration and validation"**

I'd also suggest elaborating more on the calibration and validation processes for your models. This is particularly important for the flood forecasting model, where predictive accuracy is key. How do the models perform in terms of key metrics like, i.e. root mean square error (RMSE) or skill scores against observed flood events? I'd suggest including more detail / discussion on these aspects (i.e. referencing other works).

**Response #4**

*Although we understand your suggestion to elaborate more on the calibration and validation processes, we prefer not to extend our study by detailed discussions on the calibration and validation of the model components. Firstly, all model components have been published and have undergone some calibration and validation procedures. Secondly, and more importantly, we do not propose that our specific model chain could/should be used by the insurance sector as we are aware of the limitations and uncertainties of these model components (see also our response to your comment #2). Instead, our study is intended as proof of concept, where the selection and performance of the model components are of secondary importance. We have highlighted this point even stronger in sub-section '4.3 Limitations and recommendations'.*

*"… Our explorative study has a number of limitations, some of which follow from its use of available models and datasets. Firstly, we apply models that have been developed, calibrated and validated by earlier studies for different purposes. We combine these models, or their outputs, without performing an in-depth calibration and validation of our model chain. Our study is intended as proof of concept and not as decision tool for the insurance sector. In the latter case, one would need to perform a rigorous model evaluation including sensitivity and uncertainty analysis to understand the skill of the risk forecasts…"*

**Comment #5 on "Inter-model interactions"**

Your methodology involves integrating multiple complex models. A more detailed discussion on how these models interact, particularly how uncertainties in one model propagate through to others, would significantly enhance the robustness of your approach. How do uncertainties

in the seasonal forecast model affect the accuracy of the flood frequency model and subsequently the DIFI model?

**Response #5**

*Our response to this relevant aspect of uncertainty included in our response to your comment #4.*

**Comment #6 on "Limitations"**

In the 'Recommendations' sub-section there is already a discussion on some of the limitations but, generally, they lack a bit of detail. For instance, in line 334-336: 'although both models are similar in conceptional terms, they apply different sub-models and datasets for calculating risk. Future studies should apply a more consistent model chain'; although we can go to the original papers and check what these differences are and why they are important, it would be worth including some more specificity on why their use could make the model chain somehow less 'consistent'.

**Response #6**

*We have modified the text as follows:*

*"Thirdly, the DIFI insurance model currently uses EAD data from the GLOFRIS flood risk model, following Ward et al. (2017). It then applies a yearly deviation of the EAD data based on our forecasting model, which is built on the flood risk model of Alfieri et al. (2015). Although both flood risk models are conceptually similar, future research can further improve consistency across the different (sub-) models and data by utilizing the same data for forecasting simulations."*

**Comment #7 on "Spatial and temporal coverage"**

Your study focuses on Germany, which has specific hydrological and climatic characteristics. How transferable are your findings to other geographical regions with different hydrological and climatic conditions?

**Response #7**

*We have added the following text in sub-section '4.3 Limitations and recommendations':*

*"Finally, our case study area Germany is not the best location in terms of seasonal streamflow forecasting skill. The proposed approach seems to be of even more practical use in areas outside of Europe, as other regions show more pronounced links between climate variabilities and flood characteristics (Arnal et al., 2019, Yan et al., 2020). While these limitations may reduce the value of the specific quantitative results for our case study, they do not affect the general insights gained in our study. In other words, although our results depend on the characteristics of Germany in terms of climate, hydrology, flood protection, flood-prone assets, damage processes and insurance system, the concept and methodology are transferable to other regions where they could be even more useful."*

**Technical corrections**

'Policyholders' in line 65 needs a hyphen.

*Corrected*

Figure 2 lacks the rivers (or they are barely visible, compared to figure 4).

**Response**

*We enhanced the visibility of rivers in Figure 2.*

---

## Author Comment (AC2)

*Response to Reviewer #2*

*(Comments by Reviewer #2: https://doi.org/10.5194/egusphere-2023-2862-RC2)*

We would like to thank Reviewer #2 for taking the time to review our manuscript entitled "Exploring the use of seasonal forecasts to adapt flood insurance premiums" (*https://doi.org/10.5194/egusphere-2023-2862*). We appreciate the insightful and helpful comments. Please find a point-by-point response below.

**General comments:**

Dear Authors, first thank you for your study on how seasonal flood forecasts can be integrated into the calculation of flood insurance premiums. You have done an excellent job of presenting your research and findings clearly and concisely. Overall, I believe that this manuscript makes an important contribution to our understanding of an underexplored area of flood risk management. With some minor revisions and additions, this paper has the potential to be an impactful publication in its field.

**Response:**

*We would like to thank the reviewer for the very positive general comment.*

**Specific comments (RC)**

**Comment #1**

The section detailing the methods, data used, and the explanation of the seasonal forecasting model could be expanded to enhance clarity. Specifically, information on the lead time of the forecasts, the sources of predictability, and how these elements influence the model's accuracy and reliability should be more thoroughly explained.

**Response #1**

*In the revised version of our paper, we enhanced the explanation of the methods and data in section '3 Methods and Data'. We now state the periods of the predictor and the predictand and the sources of predictability more explicitly, as follows:*

*"In this paper, we use the results of Steirou et al. (2022) for the German streamflow stations. To illustrate the idea of seasonal forecasting of flood peaks and damage, we limit our study to one of the 14 indices of Steirou et al. (2022). Specifically, we estimate the probability distribution of flood peaks in winter (Dec–Feb) where the GEV location parameter is conditioned on the precipitation in the season-ahead (Sep–Nov). Thus, the winter flood peak related to a given return period varies as function of autumn precipitation; for instance, more precipitation in autumn increases the catchment wetness, thus increasing the peak of the 200-year flood (Jongman et al., 2014). The source of predictability of the seasonal forecasting*

*model is thus the memory of the catchment which influences the antecedent conditions of flooding in the next season."*

*We have added additional text on the model's accuracy and reliability in the sub-section '4.3 Limitations and recommendations'.*

**Comment #2**

The discussion section would benefit from an expanded exploration of human drivers within the model chain.

**Response #2**

*We added the following text to sub-section '4.3 Limitations and recommendations':*

*"Fourthly, our model assumes that vulnerability is constant in time and that climate is the only driver of changes in flood hazard. It thus ignores other changes in time, such as improved flood protection and early warning systems. While such time-varying effects could be included in principle, their addition would substantially increase the uncertainty."*

**Comment #3**

Figure 2 Clarity: It is mentioned that rivers should be displayed with blue lines in Figure 2, but they are not visible.

**Response #3**

*Corrected.*

**Comment #4**

L260 – I suggest avoiding commenting on results that are not shown.

**Response #4**

*We add an appendix to show these results, i.e. maps of variability of the autumn precipitation and the variability of the observed flood peaks.*

**Comment #5**

L265 - Statistical Significance?

**Response #5**

*P-value to quantify statistical significance is added.*